# The validity and reliability of motion analysis sensor system for wheelchair users (MASSWU)

Nirawitt Suansomchit[1], Supachai Vorapojpisut[2], Sairag Saadprai[ID][1]*

1 Department of Sports Science and Sports Development, Faculty of Allied Health Sciences, Thammasat University, Pathumthani, Thailand, 2 Department of Electrical and Computer Engineering, Faculty of Engineering, Thammasat School of Engineering, Thammasat University, Pathumthani, Thailand

* sairag.saa@allied.tu.ac.th

## Abstract

The Motion Analysis Sensor System for Wheelchair Users (MASSWU) is designed to collect wheelchair movement data such as distance, duration, speed, and angular velocity during competitions and training sessions to enhance athlete performance. Developed as a simpler alternative to complex 2D motion analysis software, MASSWU was evaluated for validity, intra-rater and inter-rater reliability by comparison with the established 2D motion analysis program Kinovea. The assessment involved six wheelchair skill-related fitness tests performed under simulated real-world conditions by twenty-three healthy wheelchair users aged 18–35. The tests included One Stroke Push (distance), Muscle Power Sprint (speed), 10x5-meter Sprint (duration), Slalom (duration), and 360° Clockwise and 360° Anticlockwise Rotations (angular velocity). Validity was evaluated using the Pearson Correlation Coefficient, while intra-rater and inter-rater reliability were assessed through Intra-Class and Inter-Class Correlation Coefficients, respectively. MASSWU exhibited good to very good validity across all tests (r = 0.887–0.998). Both intra-rater reliability (ICC = 0.765–0.988) and inter-rater reliability (ICC = 0.899–0.996) ranged from good to very good. These results indicate that MASSWU is a highly valid and reliable tool compared with Kinovea for measuring wheelchair performance among these wheelchair skill-related fitness tests.

## Introduction

At present, sports training has applied knowledge of sports science and various technologies to promote training and sports competitions, such as planning the training program in advance, recording, and tracking training results, evaluating competition results, and improving training program to reduce athletes' weaknesses, etc [1]. There are not many important tools to use for developing and tracking the performance of wheelchair athletes and wheelchair users in daily life. Most sports scientists use the 2D and 3D Video System for Motion Analysis, which is a common tool used

**Data availability statement:** All relevant data are within the manuscript and its Supporting Information files.

**Funding:** We are grateful to all the participants of this study. This work was supported by the Faculty of Allied Health Sciences, Thammasat University, and by the Center of Excellence in Creative Engineering Design and Development. There was no additional external funding received for this study.

**Competing interests:** The authors have declared that no competing interests exist.

during sports training and competition to analyse movement [2]. Sports scientists will attach markers to the body positions of the athletes to be analyzed as reference points for motion and take pictures with a video camera. The program will then send data as the x, y, and z axis positions of the markers. Sports scientists will use this data to analyze and interpret motion in sports [3,4]. However, the 3D motion analysis system cannot be used in daily life because it must be done in the laboratory with a very high level of validity and reliability, and it is very expensive [5,6]. The 2D motion analysis system is more convenient for field measurement, but it still cannot analyze motion in competition or training situations throughout the entire range of motion, especially in team sports, where the athlete or wheelchair may obstruct each other's motions and prevent visual analysis at all the time. In sports stadiums or sports events that require a large area, motion analysis systems require multiple cameras installed in many locations, which is expensive and difficult to completely capture the wheelchair motion. They are also complex to place reflective markers on predetermined anatomic landmarks and require specialist for analysis [7,8].

Motion Analysis Sensor System for Wheelchair Users (MASSWU) is a sensor system to measure the motion of wheelchair users. The researchers have developed a processing system and designed a display screen for data, including duration, distance, speed, and angular velocity of wheelchair motion. It can evaluate wheelchair motion, whether in the field or with multiple players at the same time. It can process and display the analysis results immediately without having to analyse and evaluate the raw data again. It reduces the complexity of data analysis and does not require installing cameras, so that sports scientists can use it to develop and improve the efficiency of wheelchair athletes.

Therefore, the researchers would like to test the Validity, Intra-Rater, and Inter-Rater Reliability against a 2D motion analysis program (Kinovea), using six wheelchair skill-related fitness tests as the test which closely mimics the real conditions of wheelchair users.

## Materials and methods

A 21.00 x 17.80 mm board was connected to a 50.00 x 22.00 mm, 5,000-mAh battery in a 54.50 mm diameter cylindrical capsule mounted on the left wheelchair wheel axel (Fig 1).

The data acquisition and processing were performed on a XIAO nRF52840Sense board equipped with an LSM6DS3TR-C inertial measurement unit (IMU) sampling at 10 Hz. The IMU provided 6-axis linear acceleration and angular velocity data (ax, ay, az, gx, gy, gz). In this study, the primary measurement for wheel rotation was the Z-axis angular velocity from the gyroscope (gz), which corresponds to rotation around the wheel's central axis.

All signal processing and calculations were executed on the XIAO board using C/C++ code developed in the PlatformIO framework (S1 File). The raw gz signal was first passed through an exponential moving average (EMA, $\alpha = 0.8$) to attenuate high-frequency noise. The instantaneous linear velocity (v) was then computed directly from the filtered angular velocity according to equation (1), where r is the measured wheel radius:

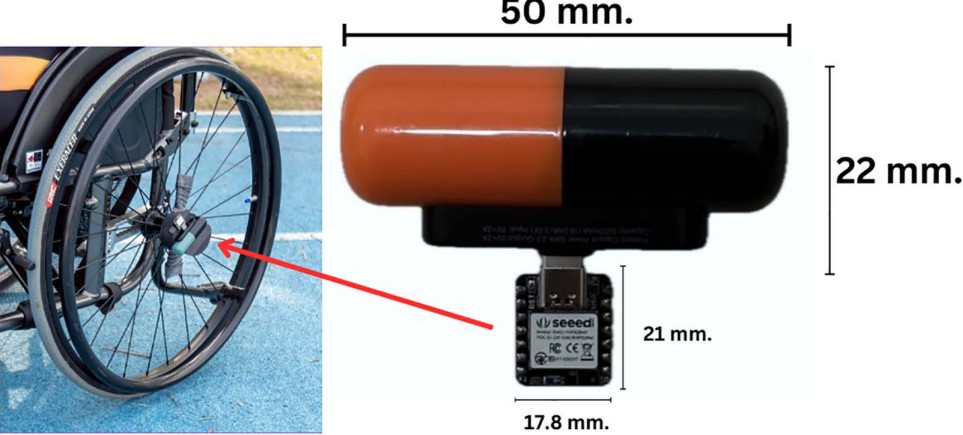

**Fig 1. XIAO nRF52840Sense Sensor and Portable Battery.**

$$v(t) = gz, filtered \cdot r \tag{1}$$

The total linear distance travelled s(t) was obtained by numerically integrating the velocity over time, as shown in equation (2), where Δt is the sampling interval:

$$s(t) = s(t-1) + v(t) \cdot \Delta t \tag{2}$$

Each trial began with a 5-second static period to establish the baseline noise level. Movement onset was detected by the XIAO board when the absolute value of the filtered gz exceeded a threshold of three standard deviations above this baseline. Once movement was detected, the onboard software continuously calculated the average linear velocity, average angular velocity, duration, and total distance. Only these processed results were transmitted to the host computer via Bluetooth Low Energy (BLE), where they were displayed in real time and saved in Excel format. The user interface supported login, function selection, data collection, and real-time graph generation for intuitive interpretation. After data collection, the system automatically analysed the results and presented them in graphical form (Fig 2) for easy visualization.

### The selection of participants

The participant selection for this research was based on specific qualifications: individuals had to be male or female, aged between 18 and 35 years, and daily wheelchair users. Participants were required to have the ability to propel their wheelchairs using their own arms and must not have had any serious underlying medical conditions such as heart disease, high blood pressure, or stroke. Additionally, participants could not suffer from severe muscle or nervous system pain that might interfere with their ability to participate in the study. The exclusion criteria included individuals scoring less than 9 on the Barthel Activities of Daily Living (ADL) scale [9], indicating limited capability for daily activities, and those who could not commit to the study period or had health concerns making physical activity unsafe, as assessed by the Physical Activity Readiness Questionnaire Plus (PAR-Q+) [10].

Participants who encounter difficulties during the research procedures—such as being unable to fully comply with the protocols or sustaining injuries during activities—are permitted to withdraw from the study at any time to safeguard their health and to maintain the integrity of the research data.

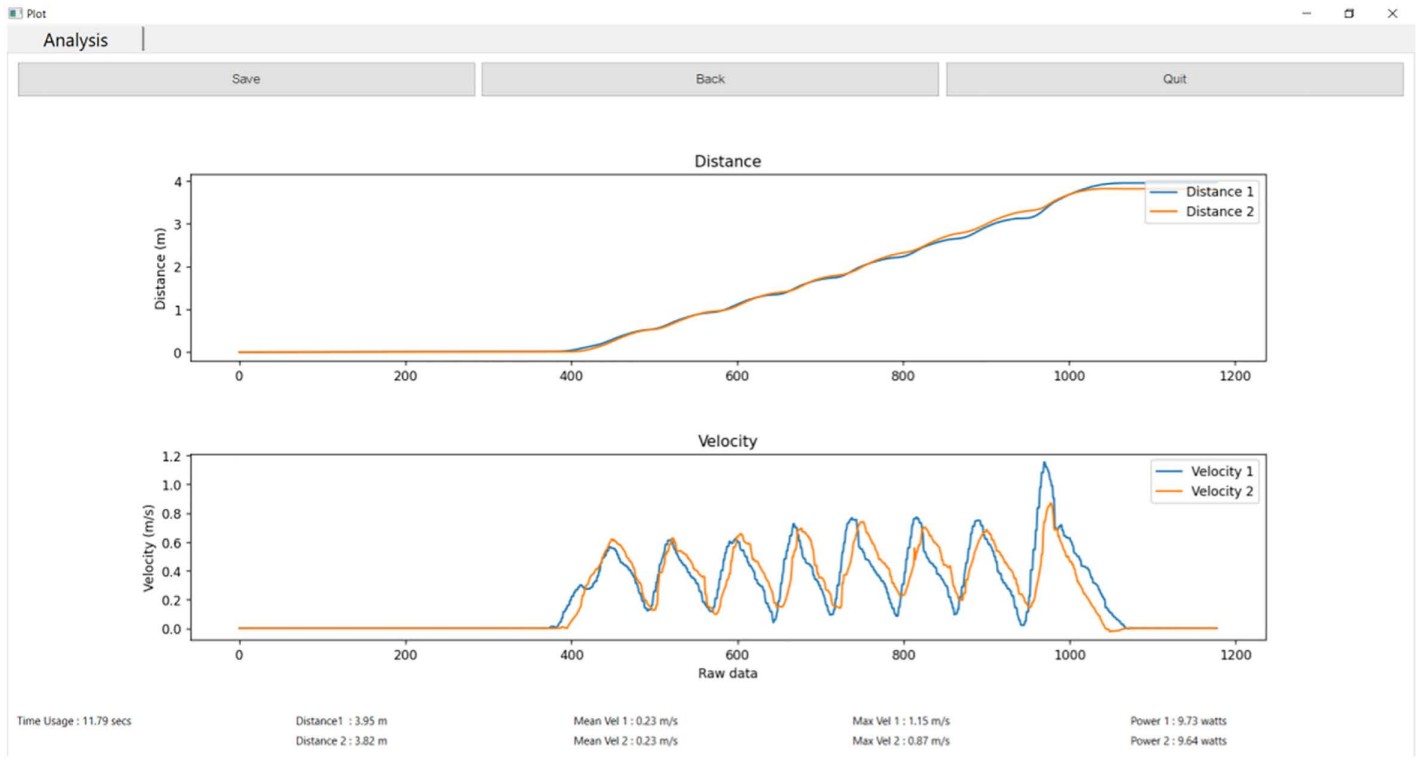

**Fig 2. Example of Data Analysis Display Screen.**

Throughout the study, the research team will closely monitor and document any potential physical or psychological risks to participants. If a participant is observed performing an incorrect posture or failing to follow the given instructions in a way that could cause harm—such as wheelchair overturning, falling from the wheelchair, or sustaining a muscle injury during testing—the researchers will immediately halt the activity to ensure maximum safety.

In the event of an injury, the researchers will assess the participant's condition and provide initial first aid. If the condition does not improve, the participant will be referred for further medical treatment at Hospital, with emergency transportation arranged as necessary. The researchers will continue to monitor the participant's condition and risk factors throughout the study to ensure both safety and peace of mind. Furthermore, participants have the right to withdraw from the research project at any time, without the need to provide a reason.

Each participant attended a single session lasting approximately three hours. Data collection took place at the indoor futsal field, Gymnasium 7, Thammasat University, Rangsit campus.

The calculation of sample size in this research used the G*Power program by setting the Effect size = 0.80, which was a large effect size according to Buchner (2010) [11]. The significance level (α err prob) was set at 0.05, and the Power value (1-β err prob) was set at 0.80. The calculation result was a minimum sample size of 15 participants. However, since the target group was vulnerable and had a high chance of drop out, the researchers therefore kept in reserve for drop out of 100%, so the number of participants was planned to 30 participants.

The study was approved by the Human Research Ethics Committee of Thammasat University (Science) (Certificate of Approval No. 080/2567) (S2 File). The researchers confirm that all methods were performed in accordance with the relevant guidelines and regulations. All participants read and signed the consent form before participating in the study.

## Methodology

The selected participants were given information about the experimental procedures, risks, and benefits, and signed in the Informed Consent Form. Then, physical readiness was assessed using the Physical Activity Readiness Questionnaire Plus (PAR-Q+). Basic information was then filled in to create an identification number for reference in the research. Basic fitness was assessed by measuring blood pressure, heart rate, and arm muscle strength using a hand grip dynamometer.

The researchers set up a video camera for the Kinovea using a Sony Cyber-shot DSC-WX800 camera. The camera was installed at a height of 1 meter and 12 meters away from the center of the test (Fig 3). After the installation of recording equipment, the participants warmed up by arm and shoulder stretches such as arm rotation, Biceps stretch, Triceps stretch, Deltoids stretch, and trunk rotation stretch. The wheelchair used in this research was the Matsunaga® (Grace Core 11b), which was adjustable for the height and width of the seat according to the user. After that, the researchers would explain and demonstrate all six-wheelchair skill-related fitness tests, starting with the One Stroke Push Test (OSPT), the Muscle Power Sprint Test (MPST), the 10x5 Meter Sprint Test (10X5MST), Slalom Test (ST), 360° Clockwise Rotation (360° CR), 360° Anticlockwise Rotation (360° AR). In this study, wheelchair skill-related fitness tests from the following credible previous research were utilized. In Validity and Reliability of Skill-Related Fitness Tests for Wheelchair-Using Youth with Spina Bifida (2017) [12] four tests validated an IMU sensor for measuring distance, speed, and duration, relevant to wheelchair users. Two additional tests from Opportunities for Measuring Wheelchair Kinematics in Match Settings; Reliability of a Three Inertial Sensor Configuration (2015) [13] confirmed the sensor's reliability in measuring angular velocity, consistent with daily wheelchair use. Moreover, the researchers controlled the variables that might affect the tests as follows: Before each testing, the condition of the field surface would be checked for consistency, to ensure there was no obstacles or damage that might affect the tire grip or uneven wear. The battery level of MASSWU would be checked to be more than 50% before and during testing to ensure that the equipment ran efficiently.

Before testing, the participants warmed up and got used to the wheelchair for 15 minutes. The participants performed 3 tests: 1 practice test and 2 real tests. The real tests were performed by 2 researchers using different sensors but were installed in the same position for each test set. The researchers will collect data from both sensors and record video footage during each test to subsequently analyse in the Kinovea software. There was a 5-minute rest between the 1st and 2nd

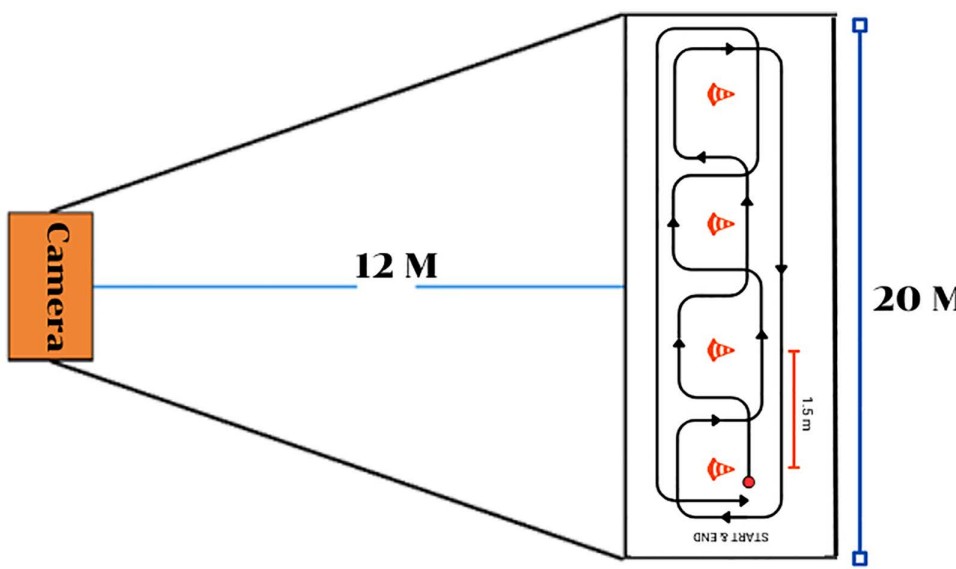

**Fig 3. Camera installation pattern.**

tests and a 10-minute rest between each test set. The heart rate must return to rest and the participants' Rating of Perceived Exertion (RPE) [14,15] must be less than or equal to 11, with the scale of 6–20 Borg's scale to assess the participants' readiness for the next test. The heart rate was set at the resting level before the test and the RPE was 11 or less. If the participants' RPE was 12 or higher, they were given an additional 2-minute rest, and if it exceeded 16, The researchers asked them to stop the next test immediately. When the participants had completed all 6 tests, the experiment was finished.

Data Collection from MASSWU and Kinovea, for the first test, data from MASSWU and Kinovea were collected from the left wheelchair axle in the side view. Data recording began when the axle passed the floor markers placed by the researchers (serving as the starting reference point) and continued until the axle came to a complete stop (serving as the ending reference point). For the second test, data were also collected from the left wheelchair axle in the side view. Recording started when the axle passed the starting reference markers and ended when it passed another set of markers positioned 15 meters away (ending reference point). For the third and fourth tests, data were collected from the left wheelchair axle in the side view, starting as it passed the floor markers. Recording ended when the right wheelchair axle in the side view passed the same position used as the starting point. For the fifth and sixth tests, data were collected from both wheel axles in the front view. Recording began when both axles were positioned at the starting floor markers and ended when both returned to the same starting position twice.

Calibration for Kinovea Analysis, before analysing each test with Kinovea, the researchers placed two cones 20 meters apart on the floor. This distance served as a calibration reference to ensure measurement accuracy, and calibration was performed before each analysis session.

## Statistical analysis

Raters measured MASSWU and Kinovea once each and averaged the results from both raters for each tool across all participants. The averages were compared using Pearson Correlation Coefficient to assess criterion-related validity. Validity was rated as low (0.00–0.50), moderate (0.50–0.74), good (0.75–0.90), or very good (0.90–1.00) [16–18]. For non-normal distributions, the Spearman Rank Test was used, and a Paired Samples t-test determined the difference between tools. Distribution normality was tested using Kolmogorof-Smirnov statistics. For non-normal distributions, the Wilcoxon Sign Rank Test was used (P<0.05).

Intra-Class Correlation Coefficient between MASSWU and Kinovea was assessed by two raters measuring twice. The first measurement from each rater was averaged, and then second measurements were averaged. The averages determined rater reliability via a Two-Way mix-effects model ICC (3,2), with significance at p<0.01. Reliability was rated as low (0.00–0.50), moderate (0.50–0.74), good (0.75–0.90), or very good (0.90–1.00) [16–18]. For non-normal distributions, the Spearman Rank Test was applied.

Inter-Class Correlation Coefficient of MASSWU was compared between raters. Two raters measured MASSWU and Kinovea twice each, and then the average of each tool was determined. The averages were analyzed using the Two-Way mix-effects model ICC (3,2). Reliability was rated as low (0.00–0.50), moderate (0.50–0.74), good (0.75–0.90), or very good (0.90–1.00) [16–18]. For non-normal distributions, the Spearman Rank Test statistic was used. And then the researchers calculate the Mean Absolute Percentage Error (MAPE) using equation (4). If MAPE is under 10% compared to Kinovea, the error is considered very low and acceptable [19].

$$MAPE = \frac{1}{\text{sample size}} \sum \left| \frac{\text{Kinovea value} - \text{MASSWU value}}{\text{Kinovea value}} \right| 100 \tag{4}$$

## Materials

In this research, the six types of skill-related fitness tests were utilized, namely:

- The One Stroke Push Test (OSPT) was a strength test for pushing the wheelchair wheels within 1 time. (Fig 4)

- The Muscle Power Sprint Test (MPST) was a speed and power test in which a wheelchair was moved forward 15 meters as fast as possible (Fig 5).

- The 10x5 Meter Sprint Test (10X5MST) was a speed and agility test in which the wheelchair was moved forward 5 meters and back and forth as fast as possible 10 times (total 50 meters) (Fig 6).

- The Slalom Test (ST) is an agility test in which the wheelchair was moved at maximum speed around four cones 1.5 meters apart (Fig 7).

- The 360° Clockwise Rotation (360° CR) and the 360° Anticlockwise Rotation (360° AR) was a coordination test of nervous system and muscles. The wheelchair would be pushed to rotate in a clockwise and counterclockwise direction using the wheels as the pivot point, starting from the wheelchair moving and rotating to a complete 360° and returning to the original position total of 2 rounds (Fig 8).

Kinovea's validity and reliability are comparable to 3D motion analysis programs [20–22] but Kinovea is better suited for field data collection [23–27]. It has served as a standard for validating devices [28–30]. Qin et al. (2023) evaluated a wearable fetal movement detection system using the NRF52840, which is the same core microcontroller as the XIAO nRF-52840Sense sensor employed in our study. Their research demonstrated that, when configured for real-time movement detection and paired with appropriate accelerometers and algorithms, this hardware platform achieved an average recognition rate and correct rate of 89.74% for detecting fetal movements in a challenging real-world scenario—specifically,

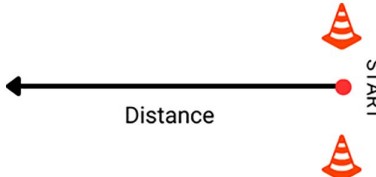

**Fig 4. The One Stroke Push Test (OSPT).**

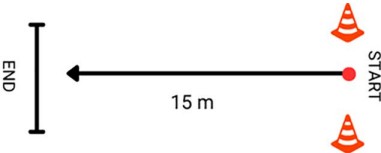

**Fig 5. The Muscle Power Sprint Test (MPST).**

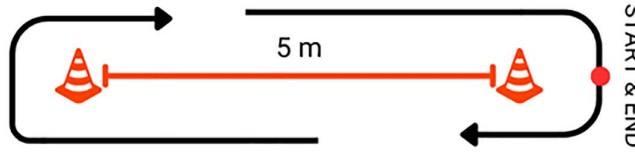

**Fig 6. The 10x5 Meter Sprint Test (10X5MST).**

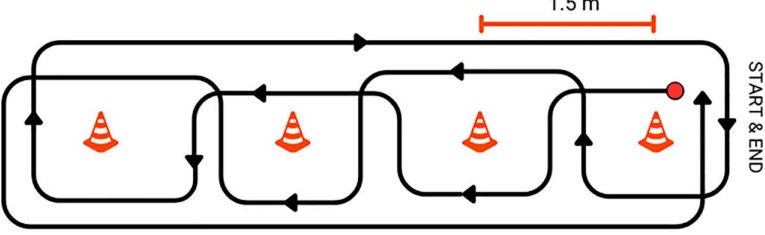

**Fig 7. The Slalom Test (ST).**

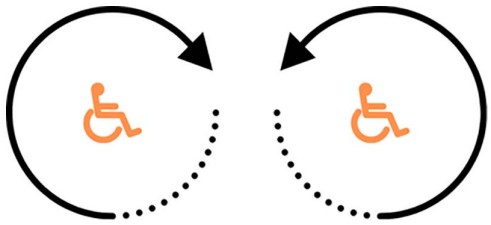

**Fig 8. The 360° Clockwise and Anticlockwise Rotation (360° CR and 360° AR).**

low-amplitude and subtle fetal motion in a clinical setting. This validation confirms the sensor's capability for accurate, reliable detection of small and complex human movements, even in environments that present high signal variability due to biological and environmental noise [31]. Although our research focuses on wheelchair movement, the technical demands are comparable-both require accurate real-time detection and differentiation of sensor signals corresponding to motion events of interest against potential background noise or artifacts. The high performance of the nRF52840-based system demonstrated by Qin et al. directly supports the suitability of using the XIAO nRF52840Sense sensor in our field setting, as it indicates the platform provides both the hardware reliability and signal fidelity necessary for precise human movement analysis.

## Results

### General characteristics of participants

The initial sample size calculation using the G*Power program recommended 15 participants. However, since the target group was vulnerable and had a high chance of drop out, the researchers therefore kept in reserve for drop out of 100%, so the number of participants was increased to 30 participants.

Though, we screened 23 individuals to ensure we would achieve sufficient sample size. After applying the inclusion/exclusion criteria and considering consent withdrawals, 23 participants remained for analysis with no drop out. A post hoc power analysis was performed with the final sample of 23, using the same parameters (effect size = 0.80, α = 0.05), which demonstrated a statistical power of 0.956. This confirms that the study was adequately powered to detect large effects as originally intended.

The participants were 23 disabled people who passed the inclusion and exclusion criteria consisted of 17 males (73.91%) and 6 females (26.09%), who signed a consent form to cooperate in the research. The participants had a wheelchair usage duration per day of 701.74 ± 347.36 minutes (Min, Max = 240, 1,140), an average weight of 65.02 ± 14.90 kilograms (Min, Max = 32, 110), an average height of 163.46 ± 16.14 centimeters (Min, Max = 120, 180), and a body mass index of 25.13 ± 10.39 kilograms/$m^2$ (Min, Max = 18.37, 71.54), and an average age of 30.04 ± 4.69 years (Min, Max = 20, 35).

## Criterion related validity of MASSWU compared with Kinovea

The MASSWU and Kinovea were measured once by two raters, and then determined the average of each tool obtained from the two raters by all participants who completed the six skill-related fitness tests. The averages of both tools were compared to examine the validity of the tools.

Regarding the distribution of data in the validity test, the researchers found that the OSPT, 10X5MST, and ST tests exhibited a normal distribution; therefore, Pearson correlation and Paired T-Test analysis was employed. In contrast, the MPST, 360° CR, and 360° AR tests demonstrated a non-normal distribution; thus, the Spearman rank test and Wilcoxon Sign Rank Test was applied.

For the tests of OSPT, MPST, 10X5MST, ST and 360° AR the researchers found that MASSWU had significant validity compared with Kinovea with a very good correlation coefficient except for the 360° CR test, with a good correlation coefficient. The researchers used a comparative statistic to determine the difference between the MASSWU and Kinovea measurements without significant difference and the percentage difference was less than 10 percent as shown in Table 1 (S3 File).

## Intra-class correlation coefficient between MASSWU and Kinovea

The reliability of MASSWU was compared with Kinovea using 2 raters, each measuring MASSWU and Kinovea. Each raters measured twice and then took the average of the 1st measurement values of each rater to determine the average and took the average of the 2nd measurement values of each rater to determine the average. Then, the average of the 1st and 2nd measurements was used to calculate the Intra-Rater Reliability.

In the analysis of data distribution for the reliability tests, it was found that the data from the OSPT and ST tests were normally distributed; therefore, the Intra-Class Correlation Coefficient (ICC) using the Two-Way Mixed-Effects Model ICC (3,2) was chosen for analysis. In contrast, the data from the 10X5MST, MPST, 360° CR, and 360° AR tests were non-normally distributed; thus, Spearman rank correlation was used.

For the OSPT, MSPT, 10X5MST, ST and 360° AR, the researchers found that the Intra-Class Correlation Coefficient (ICC 3,2) MASSWU and Kinovea were close to each other, with a very good reliability except for the 360° CR test and

**Table 1. Criterion Related Validity of MASSWU Compared with Kinovea.**

| Wheelchair Performance Tests | Devices | Mean (±SD) | % change | Paired T-Test p value | PCC r value | results |
|---|---|---|---|---|---|---|
| OSPT (m) | MASSWU | 7.574 (±2.793) | 2.837 | 0.335 | 0.955* | Very good |
| | Kinovea | 7.365 (±2.536) | | | | |
| MPST (m/s) | MASSWU | 0.506 (±0.129) | 0.393 | 0.570 | 0.982* | Very good |
| | Kinovea | 0.508 (±0.128) | | | | |
| 10X5MST (s) | MASSWU | 50.891 (±9.478) | 0.064 | 0.698 | 0.998* | Very good |
| | Kinovea | 50.924 (±9.584) | | | | |
| ST (s) | MASSWU | 24.562 (±5.149) | 0.598 | 0.491 | 0.989* | Very good |
| | Kinovea | 24.710 (±4.995) | | | | |
| 360° CR (°/s) | MASSWU | 128.328 (±26.951) | 1.556 | 0.758 | 0.887* | Good |
| | Kinovea | 126.362 (±23.797) | | | | |
| 360° AR (°/s) | MASSWU | 128.62 (±26.416) | 2.871 | 0.114 | 0.943* | Very good |
| | Kinovea | 132.422 (±27.363) | | | | |

\* Correlation is significant at the 0.01 level (Sig. 2-tailed).

\*\* Difference is significant at p < 0.05 (Sig. 2-tailed).

360° AR of MASSWU test, with a good reliability. However, the remeasurements of two raters had a difference percentage of less than 10 percent as shown in Table 2 (S3 File).

### Inter-class correlation coefficient of MASSWU and Kinovea compared between raters

The reliability of MASSWU was compared between raters using two raters who each measured MASSWU and Kinovea twice each. The average of each tool was then calculated, and the average of each tool was statistically analysed.

In the analysis of data distribution for the reliability test, it was determined that the OSPT and ST test data followed a normal distribution. Consequently, the Inter-Class Correlation Coefficient (ICC) with a Two-Way Mixed-Effects Model ICC (3,2), specifically ICC (3,2), was used. Conversely, the data from the 10X5MST, MPST, 360° CR, and 360° AR tests exhibited a non-normal distribution; therefore, the Spearman rank test was used.

For the tests of OSPT, 10X5MST, ST, 360° CR and 360° AR, it was found that the reliability of MASSWU compared between raters had a good to very good reliability correlation coefficient and the MAPE value was less than 10 percent, indicating that the error value was at a very low level, except in the MPST test, it was found that the reliability of MASSWU compared between raters had a lower reliability correlation coefficient than other tests, at a good level and the MAPE value was less than 10 percent, as shown in Table 3 (S3 File).

## Discussion

The present study demonstrated that the MASSWU system achieved good to very good validity across distance, speed, duration, and angular velocity measurements, with results strongly aligned with previous research involving similar sensor-based approaches. For distance measurement validity, the MASSWU's correlation matched those reported by Fasipe et al. (2024) [32], who utilized a commercial IMU-based system in a 6-Minute Push Test and found excellent agreement with observer-measured distances ($r = 0.970$). Speed measurement validity was also consistent with values reported by Parrington et al. (2016) [33], who validated a commercially available IMU against a LAVEG Sport laser during wheelchair sprints, obtaining a high correlation ($r = 0.920$). Regarding the validity of duration measurement, consistent with the findings reported by Coulter et al. (2011) [34], who developed an instrument to measure the physical activity of wheelchair users by assessing the motion duration of individuals with disabilities, the results demonstrated very good validity ($r = 0.981$). Moreover, Delgado-García et al. (2021) [35] reporting high validity in angular velocity measurements

**Table 2. Intra-Class Correlation Coefficient between MASSWU and Kinovea.**

| Wheelchair Performance Tests | Devices | 1st measurement Mean (SD) | 2nd measurement Mean (SD) | % Change | ICC (3,2) r value | results |
|---|---|---|---|---|---|---|
| OSPT (m) | MASSWU | 7.553 (±2.793) | 8.238 (±3.113) | 8.315 | 0.965* | Very good |
| | Kinovea | 7.382 (±2.556) | 8.195 (±2.964) | 9.920 | 0.946* | Very good |
| MPST (m/s) | MASSWU | 0.507 (±0.123) | 0.491 (±0.108) | 3.258 | 0.943* | Very good |
| | Kinovea | 0.509 (±0.126) | 0.498 (±0.115) | 2.208 | 0.939* | Very good |
| 10X5MST (s) | MASSWU | 50.957 (±9.575) | 51.265 (±11.521) | 0.600 | 0.957* | Very good |
| | Kinovea | 50.908 (±9.685) | 51.070 (±11.376) | 0.323 | 0.933* | Very good |
| ST (s) | MASSWU | 24.585 (±4.993) | 24.089 (±4.694) | 2.059 | 0.987* | Very good |
| | Kinovea | 24.695 (±5.007) | 24.090 (±4.570) | 2.511 | 0.988* | Very good |
| 360° CR (°/s) | MASSWU | 127.426 (±26.116) | 137.634 (±30.724) | 7.416 | 0.765* | Good |
| | Kinovea | 127.655 (±24.758) | 139.199 (±30.256) | 8.293 | 0.816* | Good |
| 360° AR (°/s) | MASSWU | 128.099 (±26.308) | 133.938 (±26.355) | 4.359 | 0.865* | Good |
| | Kinovea | 131.557 (±26.484) | 134.948 (±25.818) | 2.513 | 0.926* | Very good |

* Correlation is significant at the 0.01 level (Sig. 2-tailed).

**Table 3. Inter-class Correlation Coefficient of MASSWU Compared between Raters.**

| Wheelchair Performance Tests | Devices | Mean (SD) (Rater 1, Rater 2) | MAPE | r value | results |
|---|---|---|---|---|---|
| OSPT (m) | MASSWU | 7.895 (±2.924) | 3.517 | 0.985* | Very good |
| | Kinovea | 7.788 (±2.721) | | | |
| MPST (m/s) | MASSWU | 0.503 (±0.104) | 5.623 | 0.899* | Good |
| | Kinovea | 0.503 (±0.120) | | | |
| 10X5MST (s) | MASSWU | 51.111 (±10.594) | 0.615 | 0.986* | Very good |
| | Kinovea | 50.989 (±10.453) | | | |
| ST (s) | MASSWU | 24.337 (±4.810) | 1.501 | 0.996* | Very good |
| | Kinovea | 24.392 (±4.774) | | | |
| 360° CR (°/s) | MASSWU | 132.530 (±27.189) | 2.414 | 0.980* | Very good |
| | Kinovea | 133.427 (±26.433) | | | |
| 360° AR (°/s) | MASSWU | 131.018 (±25.624) | 2.239 | 0.989* | Very good |
| | Kinovea | 133.253 (±25.708) | | | |

* Correlation is significant at the 0.01 level (Sig. 2-tailed).

($r = 0.951$–$0.993$) using IMUs means these sensors accurately capture how fast the wheelchair or athlete's movements involve rotation, which is key to understanding and improving sports performance in wheelchair tennis. Compared to other validated inertial measurement units (IMUs) commonly used in sports science and rehabilitation, such as the ActiGraph GT9X, Xsens MVN, or Noraxon MyoMotion systems, MASSWU's performance is comparable in terms of reported correlation coefficients for movement parameters. For example, research using ActiGraph and Xsens devices has typically reported validity coefficients between 0.90–0.99 for distance, speed, and duration when benchmarked against reference standards during controlled wheelchair locomotion tasks [36–38]. While some advanced IMU systems may offer additional features, such as multi-segment kinematic analysis or integrated wireless synchronization for team environments, MASSWU's validity aligns with or surpasses the core measurement accuracy reported in the literature for established IMU devices in wheelchair sport and mobility monitoring contexts. Nonetheless, it is important to note that, while different IMU platforms operate based on the same principles—utilizing accelerometers and gyroscopes—they may differ considerably in key aspects such as data acquisition and analysis techniques, calibration requirements, sensor placement flexibility, cost, battery life, and overall usability. These factors should be carefully considered alongside measurement validity when selecting an IMU system for specific research or practical applications. [39,40].

In terms of Intra-Rater Reliability of distance measurement, it was found that there was a very good reliability which was consistent with research by De Vries et al. (2023) [41] found very good Intra-Rater Reliability ($r = 1.000$) when measuring wheelchair motion distance with a motion sensor versus Smartwheel and video analysis. In terms of speed measurement, it was found that there was a very good reliability which was consistent with research by Willi et al. (2024) [42] reported very good Intra-Rater Reliability ($r = 0.980$) using a motion sensor compared to an optical motion capture system for gait analysis in incomplete spinal cord injury patients during a 2-minute walk test. In terms of duration measurement, it was found that there was a very good reliability which was consistent with the research by Van der Slikke et al., 2015 [43] demonstrated very good Intra-Rater Reliability of rotation speed in Fig 8 test (ICC = 0.998) when measuring wheelchair wheel skid using a motion sensor compared to a 3D camera system. However, in terms of angular velocity measurement, in this study was found that for validity (360° CR Test) and for Intra-Class Correlation Coefficient (360° CR test and 360° AR test), MASSWU had only good validity ($r = 0.887$) and good reliability (ICC = 0.765, 0.865), possibly because the MASSWU sensor is mounted on the left wheel axle of the wheelchair to detect motion. MASSWU is a 6-axis IMU capable of measuring ax, ay, az, gx, gy, gz. The raw data from the sensor is processed to derive linear velocity and

linear displacement. MASSWU, the gz axis is primarily used, as it lies in the same plane as the wheel's rotation. When the wheelchair moves in a straight line, no centrifugal forces are involved, and tests under these conditions have shown no issues. However, in the 360° CR and 360° AR tests, the wheelchair is rotated in a circular path with the one of the wheels fixed as the pivot point and another wheel rotating clockwise and anticlockwise. This introduces centrifugal forces and motion components around the x and y axes, which contribute to measurement errors in the computed linear displacement. The computation principle is as follows: gz = angular velocity about the z-axis. Integrating gz over time yields the angular displacement about the z-axis. Multiplying this angular displacement by the wheel radius gives the linear displacement. If there is an error in orientation reconstruction from the IMU, the accuracy in assessing angular velocity will decrease, which in turn leads to the validity and reliability of the 360° CR test and 360° AR test not being at a very good level [44].

In terms of inter-rater reliability for distance measurement, the results indicated very good reliability. This finding is consistent with the research by Smith et al. (2018) [45] conducted a study evaluating the Wheelchair Skills Test specifically for power wheelchair users, assessing a total of 30 wheelchair skills. The study demonstrated excellent reliability for distance-related movement assessments, reporting a high inter-rater reliability with an ICC of 0.940. These findings support the test's consistency and precision when used to measure mobility skills involving distance and movement in power wheelchair users. In terms of the duration measurement, it was found that there was a very good reliability which was consistent with the research by Lulu Yin et al., (2025) [46] examined validity and reliability of IMUs in community-dwelling older adults in duration measurements: Consistently showed excellent agreement with gold standards for Step time test (ICCs ≥ 0.960). Also in terms of the angular velocity measurement, it was found that there was a very good reliability which was consistent with research by Cho et al. (2018) [47], in the reliability of measuring angular velocity and other kinematic parameters using an inertial measurement unit (IMU)-based system was found to be very high. Specifically, the inter-rater reliability for kinematic parameters-including those relevant to angular velocity-showed excellent consistency across measurements (ICC = 0.864 to 0.999). For instance, parameters such as hip flexion/extension, knee flexion/extension, and ankle movements—all affected by angular velocity during gait—had ICC values close to or above 0.90. This highlights the IMU system's effectiveness in reliably tracking the dynamic aspects of human movement, such as joint rotations and speeds. However, in terms of speed measurement, for the MPST was found that there was an only good reliability, although the research by Esser et al., (2012) [48] showed very good (ICC = 0.978) was found between motion sensors and OMCS for measuring CoM in Parkinson's gait. It's possibly because MASSWU calculates velocity by integrating acceleration data over time. This process can inherently introduce small, accumulated errors, such as sensor drift, which leads to a gradual deviation in measured velocity over time. As a result, the IMU's long-term accuracy in speed estimation is limited compared to video analysis methods, which track positional changes frame by frame with greater precision.

## Utilization

MASSWU could measure and analyze motion such as distance, time, speed, and angular velocity to provide data for performance assessment and design the appropriate exercise programs, including detect falls or abnormalities to provide immediate alerts and assistance. MASSWU had more advantages of convenience, portability, versatility, and privacy than cameras. The developed sensor technology would help improve the health and safety of wheelchair users and wheelchair athletes.

## Limitations and recommendations for further study

This study used MASSWU in constricted conditions, such as testing in a small arena and limited range of motion over a given area, limiting the results when applied to more complex real situations, such as rough surface or slope environments. Although the sensor used in this study had a good to very good validity and reliability in motion measurement,

there still be limitations in capturing signals in areas with many people, which might cause the sensor signal to be interfered, resulting in inaccurate test results and might not reflect the true motion. Therefore, the use of MASSWU in various environments should be studied and the system should be developed to be user-friendly, easily connect to devices and internet to increase efficiency and flexibility of use. Finally, this study utilized a sample group aged 18–35 years, characterized by high physical capacity and no underlying medical conditions, making it suitable for assessing the validity and reliability of MASSWU. However, its applicability to broader populations remains limited. To address these limitations, the researchers plan to expand the sample size and diversity in future studies to include elderly individuals, as well as people with incomplete spinal cord injuries, stroke, cerebral palsy, and other severe neurological impairments, aiming to further develop and enhance the comprehensiveness of MASSWU.

## Conclusions

This study found that the MASSWU demonstrated good to very good validity compared with Kinovea, along with good to very good intra- and inter-rater reliability. These results indicate that MASSWU can accurately detect, record, and analyze wheelchair motion parameters—such as distance, duration, speed, and angular velocity—across the six-wheelchair skill-related fitness tests (OSPT, MPST, 10×5MST, ST, 360° CR, and 360° AR) employed in this study. While these findings are promising, it should be noted that conditions in wheelchair sports competitions and training can vary considerably in terms of distance, movement patterns, coverage area, and the diversity of impairments among athletes, all of which may influence sensor performance. Therefore, any application of these findings to real-world settings should be made with caution, and further validation under competitive and training conditions, as well as in groups with different types of impairments, is recommended.

## Supporting information

**S1 File. MASSWU Codes.**
(DOCX)

**S2 File. Certificate of Approval.**
(PDF)

**S3 File. Research Data.**
(XLSX)

## Acknowledgments

We are grateful to all the participants of this study.

## Author contributions

**Conceptualization:** Supachai Vorapojpisut, Sairag Saadprai.

**Data curation:** Nirawitt Suansomchit, Supachai Vorapojpisut, Sairag Saadprai.

**Formal analysis:** Nirawitt Suansomchit, Supachai Vorapojpisut, Sairag Saadprai.

**Funding acquisition:** Supachai Vorapojpisut, Sairag Saadprai.

**Investigation:** Nirawitt Suansomchit, Supachai Vorapojpisut, Sairag Saadprai.

**Methodology:** Sairag Saadprai.

**Project administration:** Supachai Vorapojpisut, Sairag Saadprai.

**Resources:** Supachai Vorapojpisut, Sairag Saadprai.

**Software:** Supachai Vorapojpisut.

**Supervision:** Supachai Vorapojpisut, Sairag Saadprai.

**Validation:** Nirawitt Suansomchit, Sairag Saadprai.

**Visualization:** Nirawitt Suansomchit, Sairag Saadprai.

**Writing – original draft:** Nirawitt Suansomchit, Sairag Saadprai.

**Writing – review & editing:** Nirawitt Suansomchit, Supachai Vorapojpisut, Sairag Saadprai.

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
