## [Decision Letter · Decision Letter 0]

28 Jul 2025

PONE-D-25-23056The Validity and Reliability of Motion Analysis Sensor System for Wheelchair Users (MASSWU)PLOS ONE

Dear Dr. Saadprai,

Thank you for submitting your manuscript to PLOS ONE. After careful consideration, we feel that it has merit but does not fully meet PLOS ONE’s publication criteria as it currently stands. Therefore, we invite you to submit a revised version of the manuscript that addresses the points raised during the review process.

We look forward to receiving your revised manuscript.

Kind regards,

Rodrigo Rodrigues Gomes Costa, PhD

Academic Editor

PLOS ONE

“We are grateful to all the participants of this study. This work was supported by the funding under Faculty of Allied Health Sciences, Thammasat University and under Center of Excellence in Creative Engineering Design and Development.

4. Please remove your figures from within your manuscript file, leaving only the individual TIFF/EPS image files, uploaded separately. These will be automatically included in the reviewers’ PDF.

Reviewers' comments:

Reviewer's Responses to Questions

**Comments to the Author**

1. Is the manuscript technically sound, and do the data support the conclusions?

Reviewer #1: Yes

Reviewer #2: Partly

2. Has the statistical analysis been performed appropriately and rigorously? 

Reviewer #1: Yes

Reviewer #2: I Don't Know

3. Have the authors made all data underlying the findings in their manuscript fully available?

Reviewer #1: Yes

Reviewer #2: Yes

4. Is the manuscript presented in an intelligible fashion and written in standard English?

Reviewer #1: Yes

Reviewer #2: Yes

5. Review Comments to the Author

Reviewer #1: Reviewer Comments – PLOS ONE

Manuscript Evaluation

The present manuscript proposes the use of a lightweight, portable, and low-cost inertial sensor (MASSWU) for capturing kinematic metrics in wheelchair users under real-world conditions (e.g., physical exercise, functional fitness tests). The approach is timely and addresses a pertinent gap, offering a feasible alternative to traditional motion analysis systems (e.g., Vicon, Qualisys, multi-camera systems), which are typically expensive, technically demanding, and restricted to laboratory settings. The portability of the device, as well as the absence of external markers or cameras, represents a meaningful advancement in the technological accessibility and democratization of functional assessment tools.

Nevertheless, several revisions are necessary to raise the scientific rigor of the manuscript to publication standards.

Strengths

The research question is original and addresses an underexplored need in the assessment of functional performance in wheelchair users within ecologically valid environments.

The study is methodologically sound. Inclusion and exclusion criteria are clearly defined, and the selection of the six physical fitness tests is grounded in validated literature specific to wheelchair users. The data collection was performed in a controlled environment, minimizing external interferences and increasing measurement reliability.

The use of standardized and validated functional tests is an important strength of the methodology.

Statistical analyses (ICC, Pearson correlation, MAPE) are appropriate and competently conducted.

The MASSWU sensor demonstrated very high correlation with Kinovea (r > 0.95), and most ICC values exceeded 0.90, indicating excellent reliability. Additionally, the mean absolute percentage error (MAPE < 10%) for all tests supports the device’s accuracy for both clinical and athletic applications.

The study adheres to ethical standards and open data policies, in line with the journal’s expectations.

Points Requiring Revision

Scientific Writing Quality:

The manuscript suffers from poor English syntax, redundancy, and inconsistent terminology, which compromise clarity and scholarly tone. A thorough language revision by a native English speaker or a professional scientific editing service is strongly recommended.

Underperformance in the 360° Clockwise Rotation Test:

The MASSWU sensor yielded only moderate reliability in the clockwise rotation task (ICC = 0.712), in contrast to its performance in other tests. This may suggest a limitation in capturing rapid or complex rotational movements.

Recommendation: The authors should provide a more in-depth technical analysis of this limitation. Could it stem from latency in gyroscope readings? Might the single-axis reference be compromised during unidirectional rotation? A potential solution could involve repositioning the sensor or employing a second sensor to enable cross-validation of rotational data.

Comparison With Similar Technologies:

The discussion lacks a robust comparison with other validated IMU systems. A more critical contextualization of how MASSWU performs relative to existing inertial devices would strengthen the manuscript.

Sample Characteristics and Generalizability:

The participant sample includes young adults (18–35 years old) with high functional capacity and no significant comorbidities, all of whom are habitual wheelchair users. While suitable for preliminary technical validation, this limits generalizability to broader clinical populations such as individuals with incomplete spinal cord injury, stroke, cerebral palsy, or other severe neurological impairments.

Recommendation: The authors should explicitly acknowledge this limitation and expand the Discussion section to outline a concrete plan for future studies involving more clinically diverse and functionally representative cohorts.

Code and Algorithm Availability:

The analytical algorithms and processing code used for the MASSWU data were not made available. This omission limits computational reproducibility.

Recommendation: The authors are encouraged to share the code or provide supplementary material to ensure methodological transparency.

Conclusion and Recommendation

This manuscript presents significant scientific merit and contributes to the advancement of low-cost, field-applicable motion assessment tools for wheelchair users. While the findings are promising, I recommend Minor Revision prior to publication, focusing on the points outlined above, particularly improvements in scientific writing, elaboration of the sensor’s rotational limitations, and enhancement of reproducibility through code sharing.

Reviewer #2: Line 56-57, “Most sports scientists use the Video System for Motion Analysis, which is a standard method with both 2D and 3D analysis” – is it importante to cite a reference for this statement.

Line 99 - “This data is processed to calculate linear distance (based onwheelchair wheel size) and speed.“ –A more detailed description of how the data were processed would improve the understanding and potential reproducibility of the methodology used.

Line 109 – It seems that the entire paragraph are in the future, but the like would be in a project . The authors should describe how it hapends and the description of the individuals and if occurs drop outs.

Line 129 – Since the final sample size differed from the initially estimated number of participants, a post hoc power analysis should be conducted to determine the actual statistical power achieved.

Another point that requires clarification is why 27 individuals were assessed if the sample size calculation indicated that the test should be conducted with 15 participants.

Line 183 – “The data from MASSWU and Kinovea were analysed to summarize the results thoroughly and accurately

Details are needed regarding the analysis performed using Kinovea: which reference points were identified in the video for the calculation of the variables?.”

Line 212- Which variables presented a normal distribution and which did not? Following this analysis, which statistical tests were actually used for mean comparisons?

Line 213 The description of the materials would be more coherently integrated at the end of the methodology section, followed by the description of the statistical analysis.

Line 252 – “Research indicates this sensor has good validity (89.74%) in measuring fetal movement, as demonstrated in a 2023 study on wearable fetal movement detection (27)” – Explain how the results of Qin et al. (2023) could be applied or related to the context of your research.

In the tables, the data presented for the 360° CR and 360° AR tests show exactly the same means and standard deviations. This raises concern, and we would like to verify whether this is indeed correct or if there was a typographical error.

The Discussion section could better relate the findings to the existing literature by citing studies with similar results that reported high correlation coefficients, and further explore the reasons behind these findings.

452 - In the conclusion, the results may not sufficiently support such extensive extrapolation, given that conditions in wheelchair sports competitions and training are highly variable and involve greater distances and different coverage areas, which could lead to different sensor readings. Since the evaluation was conducted under specific tests, the conclusion should be limited to what was actually assessed, and any extrapolations should be made with appropriate caution."

6. PLOS authors have the option to publish the peer review history of their article (what does this mean? ). If published, this will include your full peer review and any attached files.

**Do you want your identity to be public for this peer review?** For information about this choice, including consent withdrawal, please see our Privacy Policy .

Reviewer #1: No

Reviewer #2: **Yes: ** João Veloso

---

## [Author Response · Author response to Decision Letter 1]

18 Aug 2025

All clarifications have been provided in the attached Response to Reviewers document.

---

## [Decision Letter · Decision Letter 1]

15 Sep 2025

The Validity and Reliability of Motion Analysis Sensor System for Wheelchair Users (MASSWU)

PONE-D-25-23056R1

Dear Dr. Saadprai,

We’re pleased to inform you that your manuscript has been judged scientifically suitable for publication and will be formally accepted for publication once it meets all outstanding technical requirements.

Kind regards,

Rodrigo Rodrigues Gomes Costa, PhD

Academic Editor

PLOS ONE

Reviewers' comments:

Reviewer's Responses to Questions

**Comments to the Author**

1. If the authors have adequately addressed your comments raised in a previous round of review and you feel that this manuscript is now acceptable for publication, you may indicate that here to bypass the “Comments to the Author” section, enter your conflict of interest statement in the “Confidential to Editor” section, and submit your "Accept" recommendation.

Reviewer #1: All comments have been addressed

Reviewer #2: All comments have been addressed

2. Is the manuscript technically sound, and do the data support the conclusions?

Reviewer #1: Yes

Reviewer #2: Yes

3. Has the statistical analysis been performed appropriately and rigorously? 

Reviewer #1: Yes

Reviewer #2: Yes

4. Have the authors made all data underlying the findings in their manuscript fully available?

Reviewer #1: Yes

Reviewer #2: Yes

5. Is the manuscript presented in an intelligible fashion and written in standard English?

Reviewer #1: Yes

Reviewer #2: Yes

6. Review Comments to the Author

Reviewer #1: All adjustments suggested in the first assessment were carried out appropriately and pertinently, making the study more accurate and providing more objective information.

Reviewer #2: One final suggestion: The sentence at the end of the Abstract, “These results indicate that MASSWU is a highly valid and reliable tool for measuring wheelchair performance among these wheelchair skill-related fitness”, conveys the idea that the device was validated against a gold standard. However, I am uncertain whether Kinovea could be considered as fulfilling this role. Therefore, I believe that the sentence used in the conclusion of the main body of the article would be more appropriate, with wording such as: “These results indicate that the MASSWU demonstrated good to very good validity compared with Kinovea for measuring wheelchair performance among these wheelchair skill-related fitness” — or something along those lines.

7. PLOS authors have the option to publish the peer review history of their article (what does this mean? ). If published, this will include your full peer review and any attached files.

**Do you want your identity to be public for this peer review?** For information about this choice, including consent withdrawal, please see our Privacy Policy .

Reviewer #1: No

Reviewer #2: No

---

## [Editor Report · Acceptance letter]

PONE-D-25-23056R1

PLOS ONE

Dear Dr. Saadprai,

I'm pleased to inform you that your manuscript has been deemed suitable for publication in PLOS ONE. Congratulations! Your manuscript is now being handed over to our production team.

Kind regards,

on behalf of

Professor Rodrigo Rodrigues Gomes Costa

Academic Editor

PLOS ONE